# WisecondorFF: Improved Fetal Aneuploidy Detection from Shallow WGS through Fragment Length Analysis

**DOI:** 10.3390/diagnostics12010059

**Published:** 2021-12-28

**Authors:** Tom Mokveld, Zaid Al-Ars, Erik A. Sistermans, Marcel Reinders

**Affiliations:** 1Delft Bioinformatics Lab, Delft University of Technology, Van Mourik Broekmanweg 6, 2628 XE Delft, The Netherlands; T.O.Mokveld@tudelft.nl; 2Computer Engineering, Delft University of Technology, Mekelweg 4, 2628 CD Delft, The Netherlands; Z.Al-Ars@tudelft.nl; 3Department of Human Genetics and Amsterdam Reproduction & Development Research Institute, Amsterdam UMC, Vrije Universiteit Amsterdam, Van der Boechorststraat 7, 1081 BT Amsterdam, The Netherlands; e.sistermans@amsterdamumc.nl

**Keywords:** non-invasive prenatal testing, cell-free DNA, fragment size, within-sample normalization

## Abstract

In prenatal diagnostics, NIPT screening utilizing read coverage-based profiles obtained from shallow WGS data is routinely used to detect fetal CNVs. From this same data, fragment size distributions of fetal and maternal DNA fragments can be derived, which are known to be different, and often used to infer fetal fractions. We argue that the fragment size has the potential to aid in the detection of CNVs. By integrating, in parallel, fragment size and read coverage in a within-sample normalization approach, it is possible to construct a reference set encompassing both data types. This reference then allows the detection of CNVs within queried samples, utilizing both data sources. We present a new methodology, WisecondorFF, which improves sensitivity, while maintaining specificity, relative to existing approaches. WisecondorFF increases robustness of detected CNVs, and can reliably detect even at lower fetal fractions (<2%).

## 1. Introduction

Prenatal screening is routinely used to verify and measure the health of the fetus, including the detection of chromosomal CNVs (copy number variations), in a timely manner [1]. Most prenatal screening methods are now non-invasive. Invasive methods typically lead to more conclusive results, but confer a low risk of harming the mother and/or fetus [2,3,4]. For example, verifying chromosomal aneuploidies through invasive genetic analysis carries a small, but significant, risk of causing fetal miscarriage [5]. The non-invasive methods assess fetal health through indirect means such as through morphological properties using ultrasound scans [6,7], or biochemical markers by sampling the maternal serum [8,9]. Generally, such screening methods must be performed at specific stages of the pregnancy [10], and are typically deployed in parallel [5,6], broadening the scope of detectable pathologies [10,11,12].

Since the discovery of cell-free fetal DNA (cffDNA) within the maternal peripheral bloodstream [13], it has become feasible to measure fetal DNA without invasively intervening [14]. This has opened new possibilities to safely assess fetal health utilizing genetic markers [15], and can be used to detect a wide variety of pathologies caused by events such as chromosomal aneuploidy [14,15], sub-chromosomal CNVs [16], and single-gene mutations [17]. One of the deciding factors to detect ever smaller events is largely bound by the sequencing yield, i.e., the DNA coverage, requiring more sequencing to reliably detect smaller events. The other important factor is the proportion of cffDNA that is mixed within the maternal blood, i.e., the fetal fraction, which normally contributes to 2–20% of the overall pool of available DNA [18,19], and increases slightly as the pregnancy progresses. The combination of available coverage, the fetal fraction, and the size of the CNV determine the reliability of detecting any given aneuploidy.

In clinical practice, NGS-based non-invasive prenatal screening (NIPT) with cffDNA typically utilizes shallow whole-genome sequencing (WGS) to remain economical and accessible for mass screening [20]. Such low sequencing yield leads to practical limits when detecting events, and often means that only high sensitivity and specificity is possible for larger events such as chromosomal aneuploidies. Therefore, NIPT typically tests for common trisomies, such as 21, 18, and 13 [21], since accuracy degrades too much for sub-chromosomal CNVs smaller than 5–10 megabases (Mb) [22,23,24]. Nevertheless, even for larger events, care should be taken for discordant results caused by biological phenomena such as placental mosaicism or maternal copy number variation [25,26]. While the NIPT field is dominated by methods utilizing WGS, there are also those that use RNA-seq [27], methylation profiles [28], SNPs [29], or haplotyping [30]. Each of which can fulfill an effective role in detecting specific events, especially when such data sources are integrated [27].

Most low sequencing yield NIPT methods utilize similar steps to detect events [31,32,33], starting with DNA sequencing, followed by mapping sequencing reads to a reference genome, and finally detecting whether the observed read coverage exceeds expectations based on a reference baseline. As the coverage is extremely low, the detection does not happen on a per nucleotide level, but, instead, the genome is discretized into larger equally sized regions or bins (often 250 kb to 1 Mb) in which read counts are aggregated to obtain a signal which can be compared to a baseline. Such a baseline signal, also known as a reference set, can be derived from a collection of healthy samples [34]. Although this can be effective, there are some drawbacks. For example, the experimental conditions of the sample and the healthy samples used for establishing the baseline should be identical in order to eliminate any technical biases that can confound the detection, otherwise one easily gets false positives or negatives [35]. Resequencing the baseline together with a new sample would protect towards these biases but is extremely costly.

An alternative approach to define the baseline would be to compare the observed read count with expectations derived from the sample itself. Within such a setting, a region should be compared to regions on other chromosomes that were found to behave similarly across a healthy panel. Having generated a map of similarly behaving regions from the set of healthy samples, any NIPT sample can be tested exploiting this map without (re)using the set of healthy samples. This setting is effective when the number of events is typically limited to a subset of a chromosome at a time, and is successfully adopted in clinical practice using methods such as Wisecondor and WisecondorX [35,36].

It is known that the fetal DNA fragment size is shorter than that of the maternal fragments [30]. This observation has been exploited in earlier work to predict the fetal fraction within cfDNA samples, i.e., the relative proportion of maternal and fetal DNA present within a sample [37,38]. The fragment size of cffDNA can be inferred from paired-end reads, but also by other approaches for instance, using methylation profiles [39] or based on read abundance approaches within and outside of the nucleosomes [40]. As the fetal fragment size is shorter than the maternal fragment size, this can potentially also be used to infer the abundance of fetal reads in any given genomic region. Namely, if a fetus is affected by a trisomy, an overall lower fragment size on the affected chromosome is expected as compared to the unaffected chromosomes. Our aim was to enrich the current NIPT testing procedure with this available data, as it is currently the norm to generate paired-end reads.

We introduce WisecondorFF, a methodology that detects chromosomal in the fetus from cfDNA that combines estimates based on read coverage as well as fragment size statistics. To control for variations, fragment size statistics are derived using a similar within-sample normalization approach as is exploited by current read-count procedures such as Wisecondor. We show that chromosomal CNVs can indeed be detected from inferred fragment sizes across the genome and that, when combined with read coverage, leads to improved accuracy and robustness of the NIPT procedure. As such WisecondorFF is interesting for clinical practice as the data is readily available in most clinical diagnostics facilities, as it only relies on paired-end sequencing of the DNA in maternal serum.

## 2. Materials and Methods

### 2.1. Sample Specification, Read Alignment, and Filtering

Samples were generated as part of the Dutch TRIDENT study [41]. DNA isolation, library preparation and paired end sequencing (36 bp) were performed using the Illumina VeriSeq1 sequencing protocol, according to the recommendations of the supplier (Illumina, San Diego, CA, USA). Analysis was performed by both the Veriseq algorithm (which only detects trisomies 21, 13 and 18), and by Wisecondor, which also detects other trisomies and smaller events. For this study we selected 526 samples, of which 401 had no detected chromosomal CNVs and were used as negative controls. The remaining 125 samples all tested positive for T21. The mean depth of coverage is 0.258 and 0.256, respectively, for the negative and positive samples. All read data were similarly processed and aligned to the hg19 human reference genome (excluding decoy sequences) using BWA-0.7.17 mem [42]. Paired reads are filtered according to the following criteria: (1) reads must be in the correct position/orientation for pairing; (2) only primary alignments are considered; (3) alignments must exceed a minimum mapping quality of at least 1; (4) every read should have a unique starting location (Appendix A). Alignments were compressed and left unfiltered, as all methods perform internal quality control on alignments.

### 2.2. Fragment Sizes

When quantifying the read coverage, reads are assigned to their respective region according to their starting position, whereas for read-pairs, we adopt their midpoint, i.e., the average of the starting positions. Fragment sizes were determined by the difference between the starting points of the paired reads. They were distributed with a mean of 173 bp and standard deviation of 56 bp. Fragment sizes above 300 bp were ignored as these were found to be uninformative when distinguishing between negative and positive samples. Per sample regions of interest were filtered to enable a more reliable estimate on the fragment size distribution by a lower bound on the minimal number of reads and an upper bound which is dependent on the normalized read coverage (Appendix A).

### 2.3. Preprocessing and Reference Set Construction

Read coverage is normalized across all samples. The genome is divided into regions of 5 kb and these regions were consequently scaled up to 250 kb, 500 kb, 750 kb, 1 Mb, 5 Mb, and 10 Mb to test resolution dependent differences. Region sizes smaller than 250 kb were excluded given that the read count and/or fragment size signals become too noisy at the specified sequencing yield. We did not predefine a list of genomic regions to exclude from the analysis. Instead, the within-sample methods define those based on the normalized read counts during reference set construction, with negligible differences between methods. Such uninformative regions are masked and typically appear at centromeres or highly repetitive locations where insufficient reads can be aligned. Next technical biases (e.g., GC bias) are removed by training PCAs on the negative controls (Appendix A). Note that this happens in parallel, hence two PCA mappings exist (one for each datatype), which are saved and applied to any sample that the reference is queried with.

### 2.4. WisecondorFF

WisecondorFF builds upon the same within-sample testing methodology as Wisecondor and WisecondorX [35,36]: constructing a reference set of similarly behaving regions as derived from control samples, and then processing each region in a new sample given this reference set. Hence, creating the reference set lies at the core of the methodology, which is based on the observation that the (ab)normality of a region on one chromosome can be judged relative to the behavior of similarly behaving regions (the references) in the control samples on other chromosomes. The similarity metric can be data-type dependent. For read count data, we follow Wisecondor which uses the Euclidean distance between the two vectors of read count in the two regions of consideration across the control samples. For each region, all regions on the other chromosomes are ranked based on the similarity metric, and the top *K* (here *K* = 300) regions are selected to be the reference set of regions for the region of consideration. Regions are additionally weighted according to reliability, based on the calculated distances. Doing this for all regions creates the complete reference set.

For fragment size data, the similarity metric between two regions is the Euclidean distance between the two vectors of mean fragment sizes within a region for each control sample. We experimented with different summarizations of the fragment distributions in a region, such as the median (being less predictive, Appendix A), or measures that directly capture the difference between two distributions, such as the Jensen-Shannon divergence distance (symmetrized Kullback–Leibler divergence). The latter was, however, not feasible due to the amount of noise present within the distributions (full details in Appendix A).

When testing a sample for CNVs, first, a Z-score for each (query) region and each data type is calculated separately. This Z-score can be calculated from the observed measurement (here either the read count or fragment size) in the query region with respect to the mean and standard deviation of that measurement calculated across the reference regions for the query region within that same sample. The data type specific score (of each region) can be joined into a single score by Fisher’s averaging. Next, the scores per region are used to find stretches of affected regions through segmentation. Here, we follow the methodology of WisecondorX [36] and use CBS (Circular Binary Segmentation) [43] to segment and finally obtain Z-scores of detected events. 

### 2.5. Fetal Fraction Estimation

Sample fetal fraction was estimated using SeqFF [38], this method uses a pre-trained multivariate model that was trained on the per region stratified autosomal read counts from WGS paired-end sequencing of cfDNA from maternal plasma. The average fetal fraction is 7.5% and ranges between 1.48% and 19.15% (Appendix A).

## 3. Results

### 3.1. Fragment Size Distributions Do Differ between T21 Affected and Unaffected Samples

To determine whether the fragment size can indeed be indicative of samples with a trisomy, we, first, studied the fragment size distribution within our cohort of 526 samples, across chromosome 21, for 125 samples with a trisomy of chromosome 21 as well as 401 samples having no trisomy (negative samples). Figure 1a shows that indeed, on average, a distribution shift of ~1.52 bp to shorter fragments can be observed for the T21 samples. Note that the fragment size distributions for individual samples vary considerably, likely caused by differences in fetal fraction and/or technical noise. While Figure 1a shows a shift in the chromosome-wide fragment size distribution, we can observe a similar shift in distributions when we consider smaller regions across chromosome 21, as shown in Figure 1b. These differences become, however, less noticeable when the region size decreases as fewer reads fall within a region, resulting in more noisy estimates of these distributions. With an average sequence coverage of 0.25×, we found that a minimum region size of 250 kb was required to estimate the fragment size distributions with sufficient robustness (Appendix A).

### 3.2. WisecondorFF Detects Aneuploidy Most Robustly

Within our cohort of 526 samples, we investigated the detection of common whole chromosome aneuploidies using six different approaches. The first three make use of WisecondorFF (Section 2.4), denoted as WcrFF, and detect the presence of a CNV event from read count frequencies (WcrFF^RC^), fragment size statistics (WcrFF^FS^), or both (WcrFF^RC&FS^). Two other approaches are the latest versions of Wisecondor (Wcr) [35] and WisecondorX (WcrX) [36]. Finally, we included one method that does not utilize a within sample testing approach to detect CNVs: CNVkit [44], a general purpose CNV detector, which we here use as a baseline.

Nearly all 125 T21s are detected by all methods, where we consider only events when the segments are larger than 10 Mb with Z-scores ≥ 5, Table 1:I. The performance of all methods is relatively stable across the range of selected region sizes, with exception of WcrFF^FS^, which continuously improves as region size increases, and to a lesser extent in WcrFF^RC^ and WcrX at 10 Mb, with performance sharply dropping. The former can be attributed to the increased noise in the fragment size signal, which we also encountered when attempting to call events based on the fragment size distributions of each region rather than their means (Appendix A). The latter is likely caused by the CBS algorithm used by both methods. Note that only WcrFF^RC&FS^ (at 750 kb) can detect all expected trisomies and has near optimal and stable performance within the other region sizes. The baseline, CNVkit, is competitive with the other methods, which is expected given the relative ease of detecting T21 events.

Although the majority of all expected trisomies were detected by each of the methods, this should be put into context with any additional findings, i.e., false positives. In Table 1, we also summarize the detection of events on chromosomes other than 21 in parenthesis. Generally, sensitivity increases when a smaller region size is chosen, which is especially true for Wcr. WcrFF^RC&FS^ makes far fewer false positive calls than Wcr at any resolution, while detecting slightly more than WcrX. When we change the constraints of accepting an event to segments larger than 1 Mb with |Z-score| ≥ 5, allowing smaller events to be called (Table 1:II). Wcr calls significantly more false positives (+247.17%) while still not detecting all expected T21 events, whereas WcrFF^RC&FS^ is now able to detect all expected events at nearly every resolution, with only a modest increase in false positives (+8.95%).

### 3.3. WisecondorFF Has Most Power

Next, we investigated the Z-scores generated by the different within-sample methods, since higher Z-scores indicate more power to detect an event. Per region on chromosome 21, we calculated the average Z-scores across all T21 positive samples. From Figure 2a, we can observe that the highest Z-scores are found by the WcrFF^RC&FS^ method when considering region sizes of 750 kb. Note that the Z-scores for WcrFF^RC^ drop, behaving very similarly to WcrX, and drop even more dramatically with WcrFF^FS^, which also performs worse than Wcr. Similar results are obtained across other region scales. The distribution of average Z-scores per region, as shown in Figure 2b, also display the shift towards larger Z-scores for WcrFF^RC&FS^.

### 3.4. The Power to Detect Trisomies Comes from Combining Read Count and Fragment Size Data

Next, we quantified the differences between Z-scores derived from either fragment size or read coverage as well as the combined approach. Figure 3 shows the average Z-scores across chromosome 21 for each of the 526 samples for 1 Mb sized regions for the methods. As expected, the negative samples have mean Z-scores that are closely centered around zero. Overall, we can see that the Z-score magnitude of events detected by WcrFF^RC^ are larger than those detected by WcrFF^FS^. Hence, the fragment size by itself is not as reliable or powerful as the read count. However, when combining the two measures, as in WcrFF^RC&FS^, it becomes possible to separate all negative and T21 positive samples, without detecting false positives. Additionally, the Z-score magnitude for the T21 samples is generally greater when combining both inputs compared to using only the read coverage.

### 3.5. Increased Power of WisecondorFF Allows to Detect CNVs at Lower Fetal Fractions

As a NIPT test may be performed at different stages of the pregnancy, we were interested in how the methods perform at different fractions of cffDNA available in the maternal blood plasma. Hereto, we estimated the fetal fractions of the 125 samples (Section 2.5.) and compared those to the Z-scores of detected events on chromosome 21 (Figure 4). We show that WcrFF^RC&FS^ assigns greater Z-scores in nearly all cases and at all fetal fraction ranges. Interesting is the detection of a duplication event in a sample with a sub 2% fetal fraction, which was undetectable by either Wcr or WcrX.

## 4. Discussion

Detecting CNVs in heterogeneously mixed WGS data remains challenging and is a point of ongoing research within the NIPT field, but also in adjacent fields such as cancer diagnostics from cell free tumor DNA [45]. With NIPT, one must deal with a mixture of maternal and fetal DNA in which the fetal, and thus the potentially affected part, occurs at a far smaller concentration. Moreover, when sampling at different stages of the pregnancy, different levels of fetal fraction are encountered. In addition, differences in sample preparation and sequencing can introduce sufficient noise to mask true variations within the sample. When utilizing reference-based methods that directly compare a sample signature to a reference signal, such noise is difficult to cope with. A within-sample normalization method circumvents these issues of experimental noise.

Generally speaking, NIPT methods utilizing WGS data exploit the relative frequencies of reads aligned to a reference genome. This is not without reason as the read coverage is a powerful predictor for the presence of a CNV. However, other data types are known to be informative for CNV detection. One source of information that can be readily derived from paired-end reads is the cfDNA fragment size. Although, it is general knowledge that this size differs between maternal and fetal cfDNA, in NIPT CNV detection applications this information is not utilized, even though most sequencing for NIPT nowadays is done using short paired-end sequencing.

We have shown that, both on a whole-chromosomal and a sub-chromosomal level, the fragment size distributions are indeed shifted in a detectable way when dealing with chromosomal CNVs in the fetal DNA. To detect these events, we introduced WisecondorFF a within-sample normalization method that utilizes both the relative frequencies of aligned reads as well as the fragment size to reliably detect chromosomal CNVs.

We noted that our presented method, WisecondorFF (WcrFF^RC&FS^), which utilizes both read count and fragment size statistics, make several improvements relative to other methods. Namely WisecondorFF is more sensitive than Wisecondor and WisecondorX, while being more selective, detecting fewer false positives. We found that WisecondorFF is more robust than others by assigning a higher certainty to any detected events at any fetal fraction. This leads to a benefit for WisecondorFF where calls that were previously indistinguishable from the background signal are now more likely to be detected. By relating the Z-scores of detected events with the (estimated) fetal fractions, we found that WisecondorFF can detect events at lower fetal fractions. While WisecondorFF had the best performance with a region size of 750 kb, we showed that the fragment size performs better at larger region scales. Currently, the fragment size and read coverage are integrated at the same scales. However, it would be possible to use asymmetrical region sizes across the different data types for better performance. At this time, WisecondorFF does not build gender specific reference sets, like WisecondorX does, and can, therefore, not detect CNVs on sex chromosomes. We have currently limited the use of additional information exclusively to the fragment size. However, we do believe that any type of data that can be discretized across genomic regions could potentially be integrated within a parallel within-sample normalization framework as we have shown.

By more fully exploiting the information available in current sequencing technology, we have shown that it is possible to achieve better performance within a low coverage NIPT setting by exploiting fragment size information. This opens other applications that can relate to differences in fragment size such as in cancer diagnostics or for other types of data integration within the NIPT process.

## Figures and Tables

**Figure 1 diagnostics-12-00059-f001:**
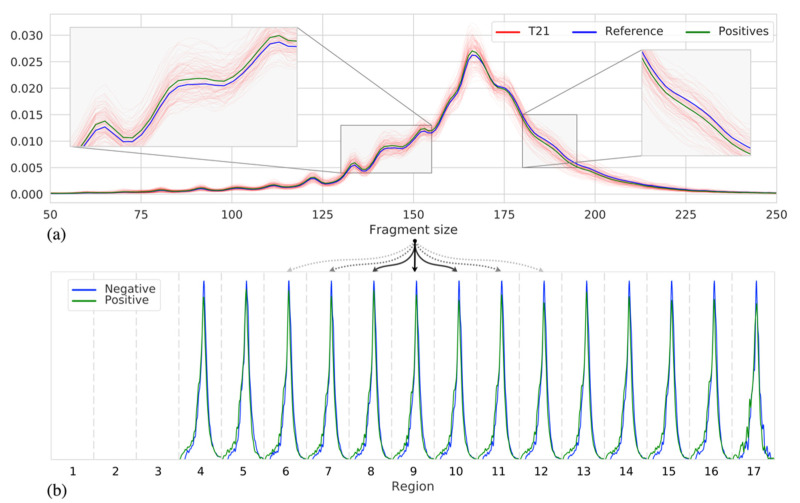
(**a**) The fragment size distributions of chromosome 21 across samples: individual T21 positive samples (red), the mean of all 125 T21 positive samples (green), and the mean of all 401 negative samples (blue). (**b**) Discretized representation of chromosome 21 into 3 Mb sized regions, per region fragment size distributions are shown (as in (**a**)) of two samples, with similar fetal fraction, one negative (blue) and one T21 positive (green).

**Figure 2 diagnostics-12-00059-f002:**
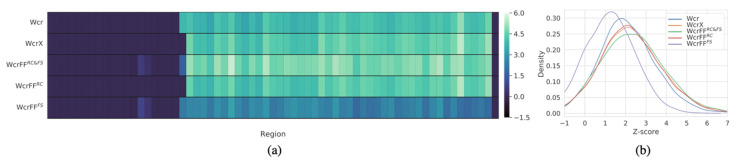
(**a**) Heatmap of the Z-scores averaged across all T21 positive samples, shown per 750 kb sized region (columns) on chromosome 21 for the different methods (rows). (**b**) Z-score distributions of all T21 positive samples on chromosome 21, with zero value Z-scores filtered out.

**Figure 3 diagnostics-12-00059-f003:**
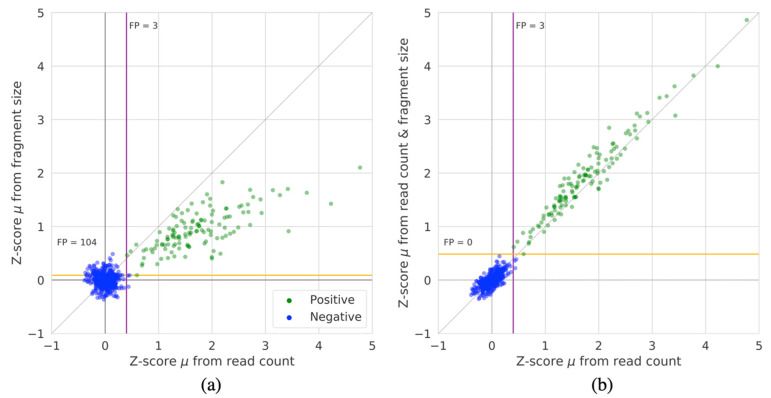
The average Z-score of all 1 Mb sized regions on chromosome 21 for all 526 samples for: (**a**) WcrFF^RC^ compared to WcrFF^FS^, and (**b**) WcrFF^RC^ compared to WcrFF^RC&FS^. The colored lines denote the Z-score cutoff boundaries that would capture all T21 positive samples for either method (purple for read count and orange for fragment size and fragment size & read count), annotations denote the number of false positives (FP) given these cut-offs.

**Figure 4 diagnostics-12-00059-f004:**
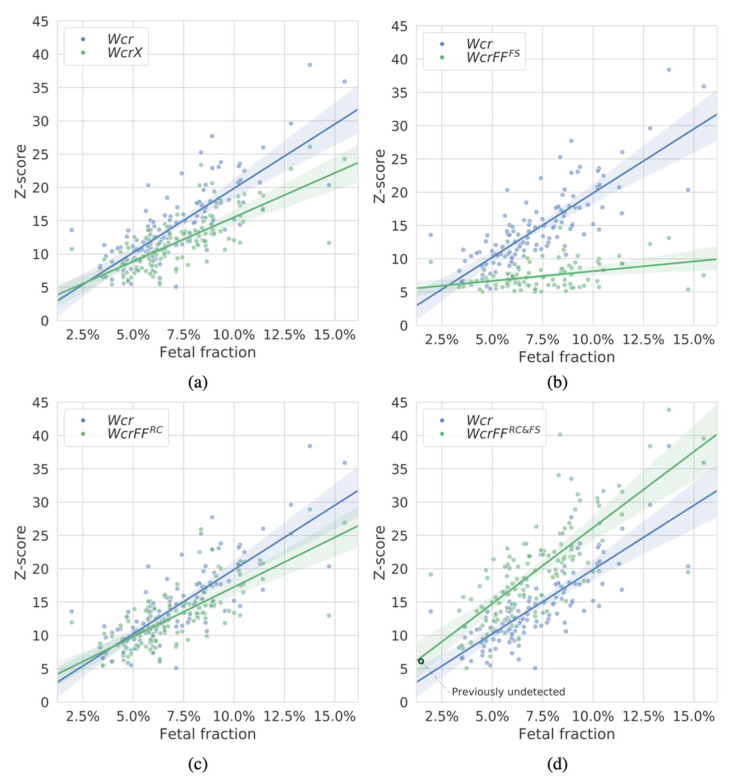
Z-scores of duplication events on chromosome 21 (an event is detected when a segment is larger than 10 Mb with a Z-score ≥ 5) as detected by the different methods in the 125 T21 samples with respect to the estimated fetal fractions of each sample. In each panel one of the methods is compared with WCR (in blue): (**a**) WcrX, (**b**) WcrFF^FS^, (**c**) WcrFF^RC^, and (**d**) WcrFF^RC&FS^ (all in green). Region size was 750 kb. Each point corresponds to an event within a sample.

**Table 1 diagnostics-12-00059-t001:** The number of detected events in the 125 T21 positive samples for the six different tested methods (rows). An event is detected when **I**: a segment is larger than 10 Mb with a Z-score ≥ 5 or **II**: a segment larger than 1 Mb with |Z-score| ≥ 5. The number of samples for which an event is detected on chromosome 21 is given for different reference region sizes (columns). Between parenthesis, we note the number events detected on one of the other chromosomes.

		250 kb	500 kb	750 kb	1 Mb	5 Mb	10 Mb
WcrFF^FS^	I	71 (0)	77 (0)	83 (0)	86 (6)	92 (14)	106 (5)
	II	71 (0)	77 (0)	83 (0)	87 (11)	92 (14)	106 (5)
WcrFF^RC^	I	122 (0)	122 (1)	122 (1)	120 (2)	121 (0)	103 (0)
	II	122 (3)	122 (2)	122 (1)	120 (2)	121 (0)	103 (0)
WcrFF^RC^^&^^FS^	I	**124 (17)**	**124** (14)	**125** (13)	**124** (14)	**124** (6)	**122** (3)
	II	125 (18)	125 (15)	125 (17)	125 (14)	124 (6)	124 (3)
Wcr	I	123 (162)	**124** (78)	123 (34)	123 (27)	122 (9)	121 (8)
	II	124 (471)	124 (285)	123 (119)	123 (215)	122 (16)	121 (8)
WcrX	I	122 (1)	122 (0)	120 (1)	121 (1)	120 (1)	103 (0)
	II	122 (4)	122 (0)	120 (1)	121 (1)	120 (1)	103 (0)
CNVkit	I	120 (375)					
	II	124 (896)					

## Data Availability

WisecondorFF is an open source tool. User-manual and software are freely available at: https://github.com/tomokveld/WisecondorFF last accessed on 29 November 2021.

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
