# Peer review of "WisecondorFF: Improved Fetal Aneuploidy Detection from Shallow WGS through Fragment Length Analysis"

_diagnostics, 2021, doi:10.3390/diagnostics12010059_

Round 1
Reviewer 1 Report
In this article the authors present an improved version of Wisecondor, a program built to identify fetal aneuploidies by massive parallel sequencing maternal cell free plasma DNA. The new version includes fragment size as a parameter to improve aneuploidy detection. Although the concept that fragment size distribution is different between the fetal/placental and maternal DNA fragments in the blood is not novel, the integration of this concept in the analysis pipeline is. Hence, this is a nice step forward in further improving cfDNA analysis.
Minor comments
- The title states ‘improved fetal aberration detection from shallow WGS through fragment length analysis’. In the article, the authors only show improvement of the methodology to detect fetal aneuploidies, more specifically trisomy 21. The term ‘fetal aberrations’ can be interpreted different ways (fetal developmental aberrations or fetal genomic aberations) but either way it would be too broad to cover the proven added value. Hence, reword to ‘aneuploidies’.
- In the results, figure 1 shows the distribution difference of fragment lengths on 21 for pregnancies with and without trisomy 21 fetuses. Could this be quantified in the text? (e.g. on average fragment length is xx nucleotides lower)?
Author Response
Reviewer 1:
R1.1. The title states ‘improved fetal aberration detection from shallow WGS through fragment length analysis’. In the article, the authors only show improvement of the methodology to detect fetal aneuploidies, more specifically trisomy 21. The term ‘fetal aberrations’ can be interpreted different ways (fetal developmental aberrations or fetal genomic aberations) but either way it would be too broad to cover the proven added value. Hence, reword to ‘aneuploidies’.
We agree with the reviewer that more precise language should be used and have carefully gone through the manuscript to correct this.
Changes to manuscript:
- Revised title to: WisecondorFF: Improved fetal aneuploidy detection from shallow WGS through fragment length analysis
- Correct for aberration usage throughout the manuscript and replace with aneuploidy/CNV.
R1.2. In the results, Figure 1 shows the distribution difference of fragment lengths on 21 for pregnancies with and without trisomy 21 fetuses. Could this be quantified in the text? (e.g. on average fragment length is xx nucleotides lower)?
We agree that it is useful to the reader to show the extent of the distribution shift in terms of nucleotide(s) given the fragment size distributions of the negative and the T21 positive samples.
Changes to manuscript:
- Added quantification of the fragment size distribution shift when comparing the reference distribution with all T21 positive samples: “Figure 1a shows that indeed, on average, a distribution shift of ~1.52 bp to shorter fragments can be observed for the T21 samples”.
Reviewer 2 Report
This paper is very interesting in that it focuses on fetal cf-DNA and shows that accurate results can be obtained with less cf-DNA in NIPT using NGS. It focuses on the fact that the length of fetal cf-DNA is shorter than that of maternal cf-DNA, and the statistical analysis takes into account not only read count but also fragment size information.
However, I think that the following part needs to be corrected.
The abstract on the paper explains the usefulness of WisecondorFF, but the Results and Discussion say that WCRFRC&FS has the highest detection rate, so I don't know which one they recommend.
↓
 Lack of explanation of abbreviations. In other words, there is no mention anywhere that WisecondorFF = WCRF RC&FS.
 Also, it is not clear whether Wisecondor [lines 82,99,111], Wisecondor(X) [lines 149,155,316] and WisecondorX [lines 177,211,327] are the same thing.
There are two figures that are numbered 3.2 and two that are numbered 3.3, which are clearly wrong.
All the figures in the PDF are small and the text is also small, so it is very difficult to understand the differences between the parts and figures that are described as excellent in the text.
The horizontal axis in Fig.1(b) and Fig.2(a) is supposed to be the physical position, i.e., Mb, but it is not described and is difficult to understand.
WCRFFS&RC in Table 1 should be unified with WCRFRC&FS in the main text.
The shaded area in Fig. 3 is difficult to understand.
Author Response
Reviewer 2:
R2.1. The abstract on the paper explains the usefulness of WisecondorFF, but the Results and Discussion say that WCRFRC&FS has the highest detection rate, so I don't know which one they recommend.
In the text we refer to WisecondorFF and WCRFRC&FS as synonymous, which we agree is confusing without stating so in the text. We have clarified this in the discussion.
Changes to manuscript:
- In the discussion we now clarify that when we discuss WisecondorFF we refer to WCRFRC&FS, i.e. the mode of WisecondorFF that integrates both read count and fragment size.
R2.2. Lack of explanation of abbreviations. In other words, there is no mention anywhere that WisecondorFF = WCRF RC&FS.
We agree with the reviewer that the abbreviations are confusing. We simplified the abbreviations such that Wisecondor, WisecondorX, and WisecondorFF are now referred to as Wcr, WcrX, and WcrFF respectively. We also emphasized in Results 3.2. that WisecondorFF is denoted as WcrFF, for which there exist three modi that either utilize read count or fragment size or both.
Changes to manuscript:
- Changed the abbreviations of Wisecondor, WisecondorX and WisecondorFF (and its three modi) throughout the text, the table, the figures, and in the supplements.
- Emphasize the abbreviation of WisecondorFF in Results 3.2.
R2.3. Also, it is not clear whether Wisecondor [lines 82,99,111], Wisecondor(X) [lines 149,155,316] and WisecondorX [lines 177,211,327] are the same thing.
While we included this “Wisecondor(X)” notation for reasons of brevity, we agree that it is confusing and have removed it.
Changes to manuscript:
- We removed mentioning of “Wisecondor(X)” throughout the manuscript and replaced it with “Wisecondor”, “WisecondorX”, or “Wisecondor and WisecondorX”.
R2.4. There are two figures that are numbered 3.2 and two that are numbered 3.3, which are clearly wrong.
We assume the reviewer referred to the numbering of the method subsections which are indeed misnumbered as stated.
Changes to manuscript:
- Corrected method subsection numbering such that they run from 3.1 to 3.5.
R2.5. All the figures in the PDF are small and the text is also small, so it is very difficult to understand the differences between the parts and figures that are described as excellent in the text.
The reviewer is correct that the figure font size is too small, which makes interpretation harder. We made changes to all manuscript figures to make the font size larger, and correct legend display.
Changes to manuscript:
- The figure font size has been increased by 50%.
- We added a legend to Figure 1a.
- Removed the legend in Figure 3b given its redundancy with respect to Figure 3a.
R2.6. The horizontal axis in Fig.1(b) and Fig.2(a) is supposed to be the physical position, i.e., Mb, but it is not described and is difficult to understand.
The reviewer is correct that the x-axi of Figures 1b and 2b refer to a physical position, which is in this instance a chromosomal region. It is important that this is understood to be the case, hence we made a number of changes to clarify this.
Changes to manuscript:
- In Figure 1b we added x-tick region numbering to make it more clear that we are referring to 3 Mb sized regions on chromosome 21. Note that we deferred from numbering the regions in Figure 2a given the number of regions shown in this heatmap.
- In the caption of Figure 1 we emphasise that what is shown are the fragment size distributions of each region of chromosome 21 if it is discretized into regions of size 3 Mb: “(b) Discretized representation of chromosome 21 into 3 Mb sized regions, per region fragment size distributions are shown (as in (a)) of two samples, with similar fetal fraction, one negative (blue) and one T21 positive (green).”.
- Clarified the caption of Figure 2 to make it more clear that Figure 2a displays 750kb sized regions on chromosome 21: “Heatmap of the Z-scores averaged across all T21 positive samples, shown per 750 kb sized region (columns) on chromosome 21 for the different methods (rows).”.
R2.7. WCRFFS&RC in Table 1 should be unified with WCRFRC&FS in the main text.
The reviewer astutely observed that the labelling was flipped around. We changed the labeling within Table 1 to reflect what is observed in the main text.
Changes to manuscript:
- Modified the labelling within Table 1 to correctly show WcrFF^RC&FS.
R2.8. The shaded area in Fig. 3 is difficult to understand.
We agree with the reviewer that the shading in Figure 3 is confusing. We removed the shadings and replaced them with colored horizontal and vertical lines to denote the cut-offs, which should be more intuitive to understand.
Changes to manuscript:
- In Figure 3 we replaced the colored shadings with colored horizontal and vertical lines.
- The caption of Figure 3 is updated to reflect these changes.